# Analgesic Effect of SKI306X on Chronic Postischemic Pain and Spinal Nerve Ligation-Induced Neuropathic Pain in Mice

**DOI:** 10.3390/biomedicines12071379

**Published:** 2024-06-21

**Authors:** Jie Quan, Chun Jing He, Ji Yeon Kim, Jin Young Lee, Chang Jae Kim, Young Jae Jeon, Chang Woo Im, Do Kyung Lee, Ji Eun Kim, Hue Jung Park

**Affiliations:** 1Department of Pain Medicine, Guizhou Provincial People’s Hospital, Guiyang 550002, China; qj890701@gmail.com (J.Q.); hcj777330@163.com (C.J.H.); 2Department of Anesthesiology and Pain Medicine, Seoul St Mary’s Hospital, College of Medicine, The Catholic University of Korea, Seoul 06591, Republic of Korea; w00dstock@naver.com (J.Y.K.); jeonyj88@naver.com (Y.J.J.); ckddn1106@naver.com (C.W.I.); dlehrud02@gmail.com (D.K.L.); jieun96@gmail.com (J.E.K.); 3Samsung Medical Center, Department of Anesthesiology and Pain Medicine, School of Medicine, Sungkyunkwan University, Seoul 06351, Republic of Korea; l7035@hanmail.net; 4Eunpyung St Mary’s Hospital, College of Medicine, The Catholic University of Korea, Seoul 06591, Republic of Korea; ksw070591@catholic.ac.kr

**Keywords:** chronic postischemic pain, glial fibrillary acidic protein, neuropathic pain, SKI306X, spinal nerve ligation

## Abstract

Neuropathic pain (NP) results from lesions or diseases affecting the peripheral or central somatosensory system. However, there are currently no drugs that are particularly effective in treating this condition. SKI306X is a blend of purified extracts of three oriental herbs (Clematis mandshurica, Trichosanthes kirilowii, and Prunella vulgaris) commonly used to treat osteoarthritis for their chondroprotective effects. Chronic postischemic pain (CPIP) and spinal nerve ligation (SNL) models were created by binding the upper left ankle of mice with an O-ring for 3 h and ligating the L5 spinal nerve, respectively. Mice with allodynia were injected intraperitoneally with 0.9% normal saline (NS group) or different doses (25, 50, or 100 mg/kg) of SKI306X (SKI groups). We assessed allodynia using von Frey filaments before injection and 30, 60, 90, 120, 180, and 240 min and 24 h after injection to confirm the antiallodynic effect of SKI306X. We also measured glial fibrillary acidic protein (GFAP) levels in the spinal cord and dorsal root ganglia to confirm the change of SKI306X administration. Both models exhibited significant mechanical allodynia. The intraperitoneal injection of SKI306X significantly increased the paw withdrawal threshold in a dose-dependent manner, as the paw withdrawal threshold was significantly increased after SKI306X administration compared with at baseline or after NS administration. GFAP levels in the SKI group decreased significantly (*p* < 0.05). Intraperitoneal administration of SKI306X dose-dependently attenuated mechanical allodynia and decreased GFAP levels, suggesting that GFAP is involved in the antiallodynic effect of SKI306X in mice with CPIP and SNL-induced NP.

## 1. Introduction

Neuropathic pain (NP) is a progressive neurological disorder associated with primary disease or dysfunction of the nervous system [1]. NP often causes severe pain and disability and impairs the quality of life and work in addition to imposing a heavy economic burden on society and families. In severe cases, it even leads to depression and suicide [2]. Due to the chronicity of various diseases, such as cancer and diabetes, the incidence of NP will further increase in the future, and NP has become an important factor leading to an increase in the global disease burden. At present, the treatment of NP mainly involves the use of drugs. According to the International Association for the Study of Pain (IASP) guidelines for NP treatment, antiepileptic drugs and tricyclic antidepressants are recommended as first-line drugs, and opioids and lidocaine patches are recommended as second-line drugs [3]. However, the long-term use of these drugs can lead to drug resistance and be accompanied by different degrees of adverse reactions, and their overall therapeutic effect is not ideal [4,5]. This may be related to the unclear pathogenesis of NP. Therefore, the treatment of NP has always been a major challenge in the medical field. Exploring effective treatment measures for NP is a major focus of medicine today that urgently needs to be undertaken, as such treatments would be extremely valuable both medically and socially.

SKI306X is the mixture of the purified extracts of three oriental herbs (*Clematis mandshurica*, *Trichosanthes kirilowii*, and *Prunella vulgaris*) [6]. In Europe and East Asia, various plants of the genus Clematis mandshurica are used to treat rheumatic pains, eye infections, fever, bone disorders, gonorrhea symptoms, gout, chronic skin diseases, varicose veins, dysentery, and malaria and as diuretics [7,8]. According to Toguibogam and as reported in The Principles and Practice of Oriental Medicine in 1610, *C. mandshurica* is effective in reducing low back and knee pain, trichosanthes can relieve fever and dry mouth, and *P. vulgaris* is effective in alleviating lymphadenitis, abscesses, and ulcers. The anti-inflammatory effects of SKI306X have been reported previously [9]. SKI306X inhibits the production of tumor necrosis factor (TNF)-α, leukotriene B4, and nitric oxide (NO) and the expression of cyclooxygenase-2 in macrophages [10]. It inhibits the production of TNF-α, prostaglandin E2, and interleukin (IL)-1β by stimulated peripheral blood (PB) monocytes [9] and is commonly used in the treatment of osteoarthritis due to its chondroprotective effects.

Pharmacological studies have shown that *C. mandshurica* has multiple beneficial biological effects, such as immunosuppressive, anti-inflammatory, antitumor, and hypoglycemic effects [11,12]. However, its analgesic effect on NP has not been documented. Therefore, we evaluated the antihyperalgesic effects of different doses of SKI306X in chronic postischemic pain (CPIP) and spinal nerve ligation (SNL) model mice.

## 2. Materials and Methods

This study was reviewed and approved by the Animal Care and Utilization Committee of the Catholic University of Korea, Seoul St. Mary’s Hospital (CUMC-2019-0248-01). The animals were treated as prescribed by the National Institutes of Health and the International Association for the Study of Pain Policy.

### 2.1. Laboratory Animals

Adult male mice C57BL/6 (6 weeks old, weighing 20–25 g) were used in the experiment. All animal experiments were performed in the semipathogen-free quarantine area of the Catholic Laboratory Animal Research Center. Five mice per cage were housed for 7 days in a humidity- and temperature-controlled (21–23 °C) environment on a 12 h light/dark cycle (starting at 7:00 a.m.) with free access to food and water. Behavioral testing and analgesia assessments were performed following ethical guidelines, and the mice were euthanized after completing the planned testing.

### 2.2. Animal Models

The mice were anesthetized with 1.5% isoflurane and 100% O_2_ as described by Coderre et al. [13]. An O-ring with an inner diameter of 5/64 inch (AS568-004) matching the size of the mouse hindlimb was placed on the left upper ankle (just above the medial malleolus) for 3 h. O-rings of the same size that were previously cut to prevent loosening were applied in the sham-operated group. After 3 h of ischemia, the small O-ring was removed to induce reperfusion, and the mice were awakened from anesthesia. SNL was performed according to the method described by Kim and Chung [14]. The mice were anesthetized with isoflurane, and the left muscles above the L4~S2 spinal cord were removed. Under a magnifying glass, the L5 transverse process was carefully resected, and the L4–L6 spinal nerves were located. The L5 spinal nerve was identified, carefully dissociated from adjacent tissues, and then tightly ligated with silk thread [15]. The wound was cleaned with disinfectant and sutured. The mice in the sham-operated group underwent the same operation under general anesthesia except for SNL. The paw withdrawal threshold (PWT) was measured the following day with von Frey filaments (Semmes–Weinstein monofilaments, Stoelting Co., Wood Dale, IL, USA). Withdrawal of the hindpaw when a filament with a bending force of less than 0.6 g was applied to the hind paw was considered to indicate mechanical allodynia.

### 2.3. Drug Administration

Mice presenting with mechanical allodynia were randomly divided into four groups. On the 16th day, the control group was given 1 mL/kg distilled water (*n* = 6). A total of three experimental groups, i.e., the SKI306X 25 mg/kg, SKI306X 50 mg/kg, and SKI306X 100 mg/kg groups (six animals in each group), received intraperitoneal injections of 25, 50, or 100 mg/kg SKI306X (Joins^®^, SK Pharma Co. Ltd., Seoul, Republic of Korea), respectively.

### 2.4. Behavioral Testing

All behavioral tests were performed at fixed times (1:00–6:00 p.m.) to avoid the effects of circadian rhythm and were conducted by the same examiner in a blinded manner. The mice were placed on a wire mesh floor in an 8 × 8 × 18 cm transparent plastic box. After the mouse was allowed to adapt to the environment for approximately 30 min, a von Frey filament (18011 Semmes-Weinstein filament, Stoelting Co., Wood Dale, IL, USA) was applied perpendicularly to the mid-plantar region of the paw for 3 s until the filament bent, and the response of the mouse was evaluated. Seven filaments with forces from 2.44 to 4.31 (0.04–2.00 g) were used [16]. The simplified up–down method described by Bonin et al. [13] was used to assess the responses of the mice in four trials starting from the trial in which the mouse started showing an avoidance response or stopped showing an avoidance response. PWT measurements were performed before drug administration and at 30, 60, 120, 180, and 240 min and 24 h after drug administration.

### 2.5. Immunohistochemistry

We measured the expression glial fibrillary acidic protein (GFAP) in the spinal cord and dorsal root ganglion (DRG) to confirm the antiallodynic effect of SKI306X. All mice with mechanical allodynia were sacrificed 60 min after the injection of SKI306X or vehicle, and the spinal cords (L4-6) and DRGs were collected. The mice injected with normal saline and the mice injected with 100 mg/kg SKI306X were anesthetized and perfused transcardially with 50 mL of 4% paraformaldehyde dissolved in 0.01 M phosphate-buffered saline (PBS) (pH 7.2–7.4). The spinal cords and DRGs of the mice were then dissected. All obtained tissues were postfixed and soaked overnight in a 30% sucrose solution. The spinal cord segments and DRGs were cut into 10 μm thick sections on a cryostat, washed three times with 0.1 M PBS for 10 min each, and incubated with 10% normal donkey serum for 1 h at room temperature. The sections were incubated with an anti-GFAP antibody (GA-5: sc-58766, mouse monoclonal, Santa Cruz Biotechnology, Dallas, TX, USA) overnight at 4 °C. The tissues were then rinsed thoroughly in PBS and incubated with Alexa Fluor 488-conjugated donkey anti-mouse secondary antibody (A21202, Thermo Fisher Scientific, Waltham, MA, USA, 1:1000) for 2 h at room temperature. After several rinses in PBS, the nuclei were stained with DAPI for 10 min, and the sections were mounted with antifade mounting medium (Vector Laboratories; Burlingame, CA, USA). Observations and image acquisition were performed using a Zeiss LSM 800 confocal microscope (Carl Zeiss Co., Ltd., Oberkochen, Germany). The mean fluorescence intensity was measured using ImageJ 2.3.0 (National Institutes of Health) and the Laboratory of Optical and Computational Instrumentation (LOCI, University of Wisconsin).

### 2.6. Statistical Analysis

All data were analyzed using GraphPad Prism v9 (GraphPad Software, Inc., San Diego, CA, USA). Data are presented as the mean ± standard error of the mean (SEM). Temporal response data are presented as the PWT to mechanical stimulation. Variables measured at different time points were compared using a repeated measures ANOVA followed by Bonferroni’s post hoc test where appropriate. *p* values less than 0.05 were considered statistically significant.

## 3. Results

### 3.1. Mechanical Allodynia in the CPIP and SNL Models

The sham operation did not cause allodynia. CPIP model mice developed marked mechanical allodynia 5 days after ischemia–reperfusion (*p* < 0.0001) (Figure 1a). SNL model mice developed marked mechanical allodynia on the third postoperative day (*p* < 0.0001) (Figure 1b).

### 3.2. Ameliorative Effect of SKI306X on Mechanical Allodynia in CPIP and SNL Model Mice

The effect of SKI306X on mechanical allodynia is shown in Figure 2. The PWT was significantly increased in the SKI306X-treated groups compared with the saline group, and the intraperitoneal injection of SKI306X increased the PWT in a dose-dependent manner. There was no significant change in the PWT in the normal saline group at any of the time points after injection. The high-dose (50 and 100 mg/kg) groups showed a more sustained increase in the PWT in response to mechanical stimulation than the SKI306X 25 mg/kg group. In CPIP model mice, 25 mg/kg SKI306X exerted an analgesic effect only at 60 min after injection (*p* < 0.0001), 50 mg/kg SKI306X had an analgesic effect from 60 to 120 min after injection (*p* < 0.0001), and 100 mg/kg SKI306X had an analgesic effect from 30 to 180 min after injection (Figure 2a) (*p* < 0.005). In SNL model mice, 25 mg/kg SKI306X exerted an analgesic effect only at 60 min after injection (*p* = 0.0093), 50 mg/kg SKI306X had an analgesic effect from 60 to 180 min after injection (*p* < 0.05), and 100 mg/kg SKI306X had an analgesic effect from 60 to 180 min after injection (*p* < 0.001) (Figure 2b).

### 3.3. GFAP

Immunohistochemistry showed that GFAP expression in the spinal cord was increased in both CPIP and SNL model mice (Figure 3b,f *p* < 0.0001) compared to control mice (Figure 3a,e). Similarly, the GFAP fluorescence intensity in the DRG was increased in CPIP and SNL model mice (Figure 4b,f) compared to control mice (Figure 4a,e) (*p* < 0.0001). The increase in GFAP expression in the spinal cord in CPIP and SNL model mice was abolished after 100 mg/kg SKI306X treatment (Figure 3c,g) (*p* = 0.005, *p* = 0.0086). Similarly, the increase in GFAP expression in the DRG in CPIP and SNL model mice was abolished after 100 mg/kg SKI306X treatment (Figure 4c,g) (*p* = 0.0037, *p* < 0.0001). Mice treated with SKI306X had a lower percentage of GFAP-positive cells (Figure 3d, *p* < 0.0001; Figure 3h, *p* < 0.0001; Figure 4d, *p* < 0.0001; Figure 4h, *p* < 0.0001).

## 4. Discussion

In some Asian countries, herbs are commonly used to treat arthritis. Although rheumatoid arthritis (RA) is not diagnosed and treated in a timely manner in many patients, some patients benefit from herbal remedies [17,18]. In the present study, SKI306X exerted antiallodynic effects and decreased GFAP expression in the spinal cord and DRG in CPIP and SNL model mice in a dose-dependent manner. Our results suggest that SKI306X exerts neuroprotective effects by reducing GFAP levers, thereby inhibiting the activation of astrocytes.

Complex regional pain syndrome (CRPS) is characterized by localized, persistent, spontaneously evoked pain that is inconsistent with the progression or magnitude of pain normally expected from trauma or lesions [19]. CRPS is generally divided into two types: CRPS type I (CRPS-I), which occurs in the absence of clear neurological damage, and CRPS type II, which accompanies neurological damage [20]. The CPIP model exhibits several vital features that mimic the clinical symptoms of CRPS-I, such as chronic thermal, mechanical, and chemical pain hypersensitivity of the affected hind limbs, followed by microvascular injury and abnormalities in regional blood flow [21,22]. During NP, inflammatory cytokine levels increase, leading to the activation of glial cells, particularly microglia and astrocytes in the spinal cord and brain. After astrocyte activation, GFAP expression increases; then, activated astrocytes release inflammatory stimuli such as cytokines and neurotrophic factors to alter the polarization of afferent neurons, and the activation of pain transmission pathways leads to central sensitization [23].

Ligation of the L5 spinal nerve, which is a highly reproducible procedure that causes minimal damage to the surrounding tissue, was performed in the current study [24]. NP symptoms produced by SNL mimic those in human patients with causalgia following nerve injury [25]. A previous study reported that NP symptoms persisted in SNL model mice for 2 months [26,27]. Surgery can also cause immediate postoperative pain and prolonged mechanical allodynia [28]. In the present study, SNL activated astrocytes and microglia at the surgical site. SNL induces spinal cord hypertrophy and increased expression of the GFAP in astrocytes and ionic calcium-binding adaptor molecule 1 (IBA1) in microglia, suggesting that these cells are activated [26]. Activated astrocytes and microglia also play a role in the initiation and maintenance of neuropathic pain after SNL [29,30]. Microglia and astrocytes in the central nervous system can release a variety of pro-inflammatory cytokines, including prostaglandin E2, tumor necrosis factor-α, IL-1, and IL-10, which are important for maintaining NP symptoms. They are the key factor that further promotes the occurrence and development of NP [31]. GFAP is a specific astrocyte marker expressed in the central nervous system. High expression of GFAP can cause damage to the central nervous system in a variety of ways [32].

SKI306X has been shown to exert chondroprotective and anti-inflammatory effects in in vitro and animal models of osteoarthritis (OA) [33]. When inflammation occurs, the hyperproliferation and abnormal activation of fibroblast-like synoviocytes (FLSs) in synovial tissue promotes the development of synovial pannus. FLSs can also secrete a variety of pro-inflammatory cytokines, such as interleukin (IL) 6, IL-1β, and tumor necrosis factor-α (TNF-α), which induce inflammatory cell infiltration and eventually lead to progressive joint destruction and dysfunction [34,35]. However, *C. mandshurica* regulates the proliferation of FLSs from rats with RA through the LncRNA OIP5-AS1/MiR-410-3p/Wnt7b signaling pathway [36]. Increased levels of pro-inflammatory cytokines such as TNF-α, IL-1β, and IL-6 and decreased expression levels of anti-inflammatory cytokines such as IL-10 are closely related to the activation of astrocytes in NP [37]. SKI306X is closely related to pain, and we speculate that this drug inhibits the activation of astrocytes through a certain pathway to exert analgesic effects.

Pro-inflammatory cytokines released by activated astrocytes play an important role in the development of NP after peripheral nerve injury [38,39]. Nerve injury or inflammation causes the activation of astrocytes in the dorsal horn of the spinal cord and satellite glial cells in the DRG, which regulates the release of pro-inflammatory cytokines, thereby increasing the excitability of neurons, and ultimately triggering pain and its maintenance [23,40]. In our study, the lowest dose of SKI306X we used was 25 mg/kg; at this dose, SKI306X exerted an antiallodynic effect, and 100 mg/kg SKI306X significantly reduced the expression of GFAP in the spinal cord and DRG. Therefore, these results suggest that SKI306X attenuates mechanical allodynia in CPIP and SNL-induced NP models by inhibiting the activation of astrocytes in the dorsal horn of the spinal cord and satellite glia in the DRG. SKI306X could be a new, safer, and more effective treatment for NP.

Limitation: However, our article still has shortcomings. In the behavioral test, we only used mechanical allodynia testing and did not further test thermal stimulation or cold stimulation. As a result, it is not clear whether this drug has an inhibitory effect on certain types of pain. In addition, this drug has not been compared with other traditional drugs that have a therapeutic effect on neuropathic pain, so the degree of pain relief is not very clear. Further animal experiments and clinical experiments are needed to confirm this.

## 5. Conclusions

This study is the first to investigate the effect of intraperitoneal injection of SKI306X on CPIP and SNL-induced NP. GFAP levels were increased in both models. Moreover, SKI306X treatment effectively decreased GFAP levels and increased the PWT, and this change coincided with a decrease in pain frequency. These results have interesting implications for further elucidating the pathological mechanisms of NP and the effect of SKI306X on GFAP.

## Figures and Tables

**Figure 1 biomedicines-12-01379-f001:**
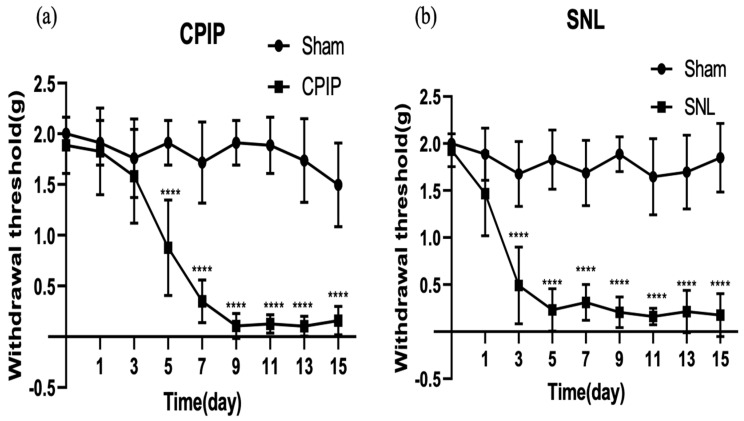
Time course of paw withdrawal responses to mechanical stimulation in CPIP and SNL model mice. (**a**): Paw withdrawal threshold of the sham group and CPIP; (**b**): Paw withdrawal threshold of the sham group and CPIP. The results are expressed as the mean ± standard error of the mean (n = 6 per group). **** *p* < 0.0001, significant difference compared with the normal saline group. CPIP: chronic postischemic pain; SNL: spinal nerve ligation.

**Figure 2 biomedicines-12-01379-f002:**
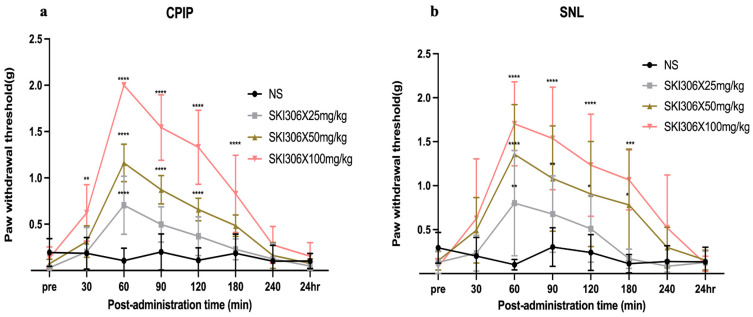
Effect of SKI306X on mechanical allodynia. (**a**): Paw withdrawal threshold within 24 h after SKI306s injection in CPIP group; (**b**): Paw withdrawal threshold within 24 h after SKI306s injection in SNL group. The PWT was measured before and after the intraperitoneal injection of normal saline, 25 mg/kg SKI306X, 50 mg/kg SKI306X, or 100 mg/kg SKI306X 100. The results are expressed as the mean ± standard error of the mean (6 in each group). SKI306X had the strongest alleviating effect on mechanical allodynia at a dose of 100 mg/kg. * *p* < 0.05, ** *p* < 0.005, *** *p* < 0.001, **** *p* < 0.0001; compared with the normal saline group at each time point.

**Figure 3 biomedicines-12-01379-f003:**
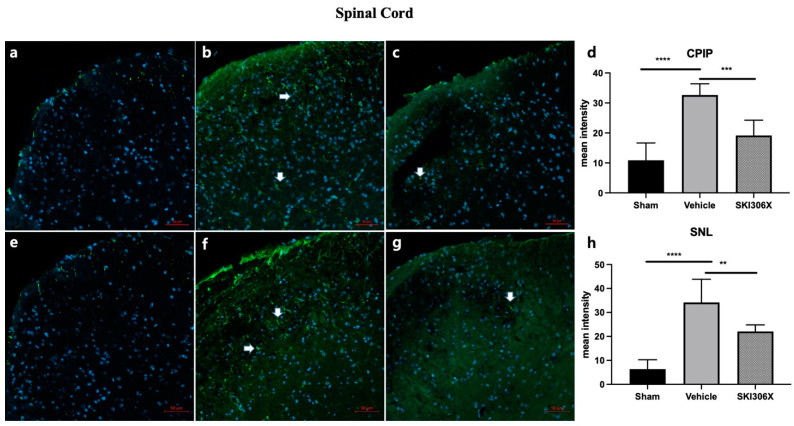
Effects of SKI306X on GFAP expression in the spinal cord in CPIP and SNL model mice. Arrows indicate astrocytes. The immunohistochemical staining results show that GFAP expression in the spinal cord was increased in CPIP and SNL model mice. (**a**) Representative image of GFAP staining in the spinal cord in sham-operated mice. (**b**) Typical image of GFAP staining in the spinal cord in CPIP model mice. (**c**) Representative image of GFAP staining in the spinal cord in CPIP model mice treated with 100 mg/kg SKI306X. (**d**) The percentage of GFAP-positive cells in the spinal cord was significantly lower in CPIP model mice injected with 100 mg/kg SKI306X than in untreated CPIP model mice. (**e**) Typical image of GFAP staining in the spinal cord in sham-operated mice. (**f**) Representative image of GFAP staining in the spinal cord in SNL model mice. (**g**) Typical image of GFAP staining in the spinal cord in SNL model mice after treatment with 100 mg/kg SKI306X. (**h**) The percentage of GFAP-positive cells in the spinal cord was significantly lower in SNL model mice injected with 100 mg/kg SKI306X than in untreated SNL model mice (n = 6 per group). GFAP: glial fibrillary acidic protein; CPIP: chronic postischemic pain; SNL: spinal nerve ligation. ** *p* < 0.005, *** *p* < 0.001, **** *p* < 0.0001.

**Figure 4 biomedicines-12-01379-f004:**
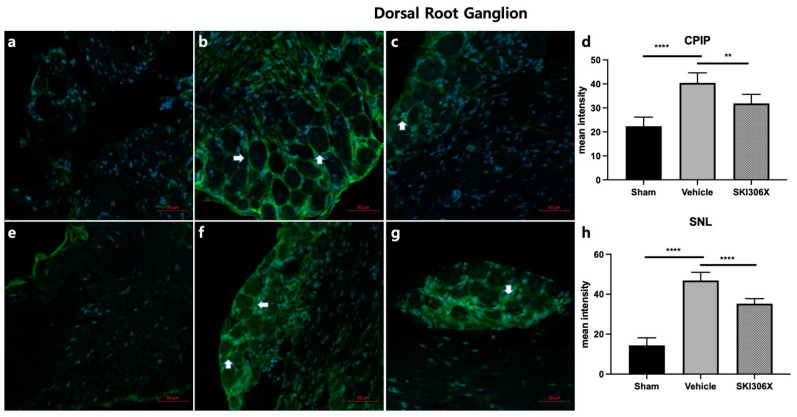
Effects of SKI306X on GFAP expression in the DRG of CPIP and SNL model mice. Arrows indicate satellite glial cells. The immunohistochemical staining results show that GFAP expression in the DRG was increased in CPIP and SNL model mice. (**a**) Representative image of GFAP staining in the DRG in sham-operated mice. (**b**) Typical image of GFAP staining in the DRG in CPIP model mice. (**c**) Representative image of GFAP staining in the DRG in CPIP model mice after treatment with 100 mg/kg SKI306X. (**d**) The percentage of GFAP-positive cells in the DRG was significantly lower in CPIP model mice injected with 100 mg/kg SKI306X than in untreated CPIP model mice. (**e**) Typical image of GFAP staining in the DRG in sham-operated mice. (**f**) Representative image of GFAP staining in the DRG in SNL model mice. (**g**) Typical image of GFAP staining in the DRG in SNL model mice after treatment with 100 mg/kg SKI306X. (**h**) The percentage of GFAP-positive cells in the DRG was significantly lower in SNL model mice injected with 100 mg/kg SKI306X than in untreated SNL model mice (n = 6 per group). GFAP: glial fibrillary acidic protein; CPIP: chronic postischemic pain; SNL: spinal nerve ligation; DRG: dorsal root ganglia. ** *p* < 0.005, **** *p* < 0.0001.

## Data Availability

The data presented in this study are available on request from the corresponding author.

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
