# Peer review of "Analgesic Effect of SKI306X on Chronic Postischemic Pain and Spinal Nerve Ligation-Induced Neuropathic Pain in Mice"

_biomedicines, 2024, doi:10.3390/biomedicines12071379_

Round 1

Reviewer 1 Report

Comments and Suggestions for Authors

In the Abstract the authors wrote the sentence "We also measured glial fibrillary acidic protein (GFAP) levels in the spinal cord and dorsal root ganglia to confirm the antiallodynic effect of SKI306X." It seems not correct, GFAP staining, so the response of glial cells, is not a direct measure of allodynia but a frequent response to pain condition. So it should be correctly rephrased, my suggestion is to firstly describe behavioral results and then move to the ex vivo effects. The glial response is an alteration that represent a maladaptive response related to pain (see doi: 10.1016/j.expneurol.2014.06.016.), the reduction induced by a treatment is a sign of neuroprotection. This consideration should be applied also throughout the manuscript.

- The alteration of pain response induced by the model is well represented in figure 1. But I am not able to see in the manuscript when (which day) the tested compound was administered to evaluate its pain relieving profile. Please clarify.

- Only von frey was used to evaluate pain threshold. It is important to add another test (thermal stimulus e.g.)

- A positive compound, clinically relevant (e.g. duloxetine or pregabalin for neuropathic pain) should be added to allow a comparison of the effect of the SKI306X

Comments on the Quality of English Language

Minor editing is necessary

Author Response

  1. In the Abstract the authors wrote the sentence "We also measured glial fibrillary acidic protein (GFAP) levels in the spinal cord and dorsal root ganglia to confirm the antiallodynic effect of SKI306X." It seems not correct, GFAP staining, so the response of glial cells, is not a direct measure of allodynia but a frequent response to pain condition. So it should be correctly rephrased, my suggestion is to firstly describe behavioral results and then move to the ex vivo effects. The glial response is an alteration that represent a maladaptive response related to pain (see doi: 10.1016/j.expneurol.2014.06.016.), the reduction induced by a treatment is a sign of neuroprotection. This consideration should be applied also throughout the manuscript.

Response: What the reviewer said makes sense, but many kinds of literature have also confirmed that intrathecal administration can produce anti-allodynia and reduce GFAP in the dorsal spinal cord and dorsal root ganglia (doi: 10.1016/j.brainres.2021.147721;doi: 10.3344/kjp.2020.33.1.23.)

  1. The alteration of pain response induced by the model is well represented in figure 1. But I am not able to see in the manuscript when (which day) the tested compound was administered to evaluate its pain relieving profile. Please clarify.

Response: Generally, we give compounds after the formation of mechanical allodynia, that is, on the 16th day.It has been modified.Please see the attachment.

  1. Only von frey was used to evaluate pain threshold. It is important to add another test (thermal stimulus e.g.)

Response: The reviewer's suggestion is very reasonable. Next time we will pay attention to adding thermal stimulation.

  1. A positive compound, clinically relevant (e.g. duloxetine or pregabalin for neuropathic pain) should be added to allow a comparison of the effect of the SKI306X

Response: Thank you for your suggestion, based on the literature records we checked

Pregabalin single injection dose 30-120mg/kg; Althobaiti YS, Almalki A, Alsaab H, Alsanie W, Gaber A, Alhadidi Q, Hardy AMG, Nasr A, Alzahrani O, Stary CM, Shah ZA. Pregabalin: Potential for Addiction and a Possible Glutamatergic Mechanism. Sci Rep. 2019 Oct 22;9(1):15136.

Duloxetine single injection dose 15,30mg/kg; Tawfik MK, Helmy SA, Badran DI, Zaitone SA. Neuroprotective effect of duloxetine in a mouse model of diabetic neuropathy: Role of glia suppressing mechanisms. Life Sci. 2018 Jul 15;205:113 -124.

This article mainly wants to highlight whether SKI306X has a therapeutic effect on neuropathic pain, and at which dose the effect is optimal. Therefore, we will pay attention to these details when doing research on neuropathic pain. Thank you for your advice.

Reviewer 2 Report

Comments and Suggestions for Authors

The reviewer declare no conflict of interest with the authors and their affiliated institutions.

1. The abstract should include an introductory sentence on what is SKI306X. The readers wouldnt know.

2. The strain of laboratory mice used in this study should be specified.

3. The description on the von Frey filamaent protocol should be improved. How many measurements were taken at each timepoint?

4. Kindly elaborate on how the immunofluorescnet intensity was measured? How does the author ensure that the quantified field among all IF immunogrammes were consistent?

5. Line 147, "Data are presented as the mean ± standard deviation of the mean (SEM)". Please be clear, SD or SEM?

6. all immunogrammes lacks scale to indicate the unit of magnification. 

7. Why in the 1st paragraph of disucssion, the author elaborate on rheumatoid arthritis? the study is on neuropathic pain....

"

Author Response

  1. The abstract should include an introductory sentence on what is SKI306X. The readers wouldnt know.

Response: Thank you for your suggestion, the text has been modified

 SKI306X is a blend of purified extracts of three oriental herbs (Clematis mandshurica, Trichosanthes kirilowii, and Prunella vulgaris) commonly used to treat osteoarthritis for their chondroprotective effects"Please see the attachment."

2.The strain of laboratory mice used in this study should be specified.

Response: C57BL/6.already edited"Please see the attachment."

  1. The description on the von Frey filamaent protocol should be improved. How many measurements were taken at each timepoint?

Response: Already edited "Please see the attachment."

4.Kindly elaborate on how the immunofluorescnet intensity was measured? How does the author ensure that the quantified field among all IF immunogrammes were consistent?

Response: 1. After adding the image, extract a single channel (Image-Color-Split Channels)

  1. Adjust the threshold and select the appropriate area (Image-Adjust-Threshold). The essence of setting thresholds is to select as many signals as possible and not select the background.
  2. Select the appropriate threshold algorithm (Image-Adjust-Auto Threshold)
  3. Set the parameters that need to be measured (Analyze-Set Measurements), and confirm that the Mean gray value and Limit to the threshold are checked (very important)
  4. Analyze-Measure

When taking photos with confocal, set the same parameters to ensure the quantization field is consistent.

  1. Line 147, "Data are presented as the mean ± standard deviation of the mean (SEM)". Please be clear, SD or SEM?

Response: standard error of the mean (SEM)   Already edited "Please see the attachment."

  1. all immunogrammes lacks scale to indicate the unit of magnification.

.

Response: All immunofluorescence images have been marked with a 200x scale bar in the lower right corner of the original image, followed by a 50um scale bar.

  1. Why in the 1st paragraph of disucssion, the author elaborate on rheumatoid arthritis? the study is on neuropathic pain.

Response: This question is fascinating. The purpose of elaborating on rheumatoid arthritis in this section is that the mechanism of the analgesic effect of some Chinese herbal medicines on rheumatoid arthritis is very similar to the pain mechanism of neuropathic pain. Therefore, the compounds in this article have an analgesic effect on rheumatoid joints. It has been proven to be effective. This part is also mentioned in the introduction section, so I want to emphasize that because of this principle, we speculate whether this compound is effective for neuropathic pain.

Round 2

Reviewer 1 Report

Comments and Suggestions for Authors

Point 1

Response of the Authors:

Response: What the reviewer said makes sense, but many kinds of literature have also confirmed that intrathecal administration can produce anti-allodynia and reduce GFAP in the dorsal spinal cord and dorsal root ganglia (doi: 10.1016/j.brainres.2021.147721;doi: 10.3344/kjp.2020.33.1.23.)

Referee: Perfectly agree but it does not mean that the regulation of GFAP is a measure of allodynia. So please correct the manuscript according to my suggestion

Point 2 – OK

Point 3

Response: Response: The reviewer's suggestion is very reasonable. Next time we will pay attention to adding thermal stimulation.

Referee: The authors did  not perform new experiments, so the measurement of pain by a single test remains limited. Please modify the statement on pain relief induced by the compound and clearly state that these are preliminar results obtained by a single analysis only

Point 4. Again the authors skipped the request. My request was to compare the effect of their product with a clinically used compound. This was not done during the revision, so the effect of the compoun remains isolated. Again, at least clearly state that the effect of the new compound was not compared with reference drugs so a comparison is not possible.

Author Response

May 28, 2024

Journal of Biomedicines

Manuscript revision biomedicines-2992198-R2 "Analgesic effect of SKI306X on chronic postischemic pain and spinal nerve ligation-induced neuropathic pain in mice"

Dear Academic Editor and Reviewer,

Thank you so much for the comments on our manuscript. We have revised our manuscript based on your notes. We hereby submit a copy of the revised paper electronically.

We are also sending a revision letter describing our responses to comments.

We would very appreciate if you could review this paper again for publication in Journal of Biomedicines.

Sincerely yours,

Hue Jung Park

Manuscript ID: biomedicines-2992198-R2

Point-by-Point response

<Reviewer #1> (Round 2)

  1. Referee: Perfectly agree but it does not mean that the regulation of GFAP is a measure of allodynia. So please correct the manuscript according to my suggestion

Response: I totally agree with your opinion. As you suggested, we modified the manuscript. Indicated in red in the text.

We also measured glial fibrillary acidic protein (GFAP) levels in the spinal cord and dorsal root ganglia to change before and after SKI306X administration were confirmed.

  1. Referee: The authors did not perform new experiments, so the measurement of pain by a single test remains limited. Please modify the statement on pain relief induced by the compound and clearly state that these are preliminar results obtained by a single analysis only

Response: I agree with you. Unfortunately, we were unable to conduct further experiments. The first author, Jie Quan, conducted the experiment while in Korea, but has now moved to China, making further experiments impossible. We hope you understand this. We only performed mechanical allodynia testing and did not further test thermal stimulation or cold stimulation. Regarding this, as the reviewer pointed out, the pain reduction effect of SKI306X will be described as preliminary results obtained only through mechanical allodynia testing. Indicated in red in the text.

  1. Point 4. Again the authors skipped the request. My request was to compare the effect of their product with a clinically used compound. This was not done during the revision, so the effect of the compoun remains isolated. Again, at least clearly state that the effect of the new compound was not compared with reference drugs so a comparison is not possible.

Response: Thanks for your good comments. As the reviewer mentioned, it would have been a better study if the experiment had been compared with A-positive compounds such as duloxetine or pregabalin. To date, there has been no study comparing SKI306X or its components Clematis mandshurica, Trichosanthes kirilowii, and Prunella vulgaris with duloxetine or pregabalin. Unfortunately, we were unable to conduct a comparative study with A positive compound, and we will do a comparative study in the next experiment. This will be described as a limitation of our research. Indicated in red in the text.

Reviewer 2 Report

Comments and Suggestions for Authors

no further comments for my part.

Author Response

Thank you very much for your correction

Round 3

Reviewer 1 Report

Comments and Suggestions for Authors

The authors well respond to my comments but I kindly invite them to insist in modifying the text accordingly. 

Please revise the language, e.g. in the abstract the new sentences are not clear 

"We also measured glial fibrillary acidic protein (GFAP) levels in the spinal cord and dorsal root ganglia to change before and after SKI306X administration were confirmed." Please rephrase and better explain

Other statement are not correct, in the discussion "Our results suggest that SKI306X inhibits the activation of astrocytes and microglia by reducing the frequency of pain via GFAP suppression." 

"by reducing the frequency of pain" what does it measn? 

"via GFAP suppression": there is not evidence for a direct link between pain decrease and GFAP reduction (and not suppresion)

Please carefully revise all the manuscript

Comments on the Quality of English Language

To improve clarity

Author Response

June 1, 2024

Journal of Biomedicines

Manuscript revision biomedicines-2992198-R3 "Analgesic effect of SKI306X on chronic postischemic pain and spinal nerve ligation-induced neuropathic pain in mice"

Dear Academic Editor and Reviewer,

Thank you so much for the comments on our manuscript. We have revised our manuscript based on your notes. We hereby submit a copy of the revised paper electronically.

We are also sending a revision letter describing our responses to comments.

We would very appreciate if you could review this paper again for publication in Journal of Biomedicines.

Sincerely yours,

Hue Jung Park

Manuscript ID: biomedicines-2992198-R3

Point-by-Point response

  1. Please revise the language, e.g. in the abstract the new sentences are not clear 

“We also measured glial fibrillary acidic protein (GFAP) levels in the spinal cord and dorsal root ganglia to change before and after SKI306X administration were confirmed." Please rephrase and better explain

Response: We also measured glial fibrillary acidic protein (GFAP) levels in the spinal cord and dorsal root ganglia to confirm the change of SKI306X administration

  1. Other statement are not correct, in the discussion "Our results suggest that SKI306X inhibits the activation of astrocytes and microglia by reducing the frequency of pain via GFAP suppression." 

by reducing the frequency of pain" what does it measn? 

"via GFAP suppression": there is not evidence for a direct link between pain decrease and GFAP reduction (and not suppresion)

Please carefully revise all the manuscript

Response: Our results suggest that SKI306X exerts neuroprotective effects by reducing GFAP levels, thereby inhibiting the activation of astrocytes .
